# Electrochemical Adsorption on Pt Nanoparticles in Alkaline Solution Observed Using In Situ High Energy Resolution X-ray Absorption Spectroscopy

**DOI:** 10.3390/nano9040642

**Published:** 2019-04-20

**Authors:** Shogo Kusano, Daiju Matsumura, Kenji Ishii, Hirohisa Tanaka, Jun’ichiro Mizuki

**Affiliations:** 1Graduate School of Science and Technology, Kwansei Gakuin University, 2-1 Gakuen, Sanda, Hyogo 669-1337, Japan; mahilu12h01m@kwansei.ac.jp (S.K.); hirohisa.tanaka@kwansei.ac.jp (H.T.); 2Materials Sciences Research Center, Japan Atomic Energy Agency, 1-1-1 Koto, Sayo, Hyogo 679-5148, Japan; daiju@spring8.or.jp; 3Synchrotron Radiation Research Center, National Institutes for Quantum and Radiological Science and Technology, 1-1-1 Kouto, Sayo, Hyogo 679-5148, Japan; kenji@spring8.or.jp

**Keywords:** ORR, carbon-supported platinum nanoparticles, superoxide anion, alkaline solution, Δ*μ* XANES, HERFD-XAS

## Abstract

The oxygen reduction reaction (ORR) on Pt/C in alkaline solution was studied by in situ high energy resolution X-ray absorption spectroscopy. To discuss the X-ray absorption near-edge structure (XANES), this paper introduced the rate of change of the Δ*μ* (RCD), which is an analysis method that is sensitive to surface adsorption. The surface adsorptions as hydrogen (below 0.34 V), superoxide anion (from 0.34 V to 0.74 V), hydroxyl species (from 0.44 V to 0.74 V), atomic oxygen (above 0.74 V), and α-PtO_2_ (above 0.94 V) were distinguished. It is clarified that the catalytic activity in an alkaline solution is enhanced by the stability of atomic oxygen and the low stability of superoxide anion/peroxide adsorption on the platinum surface.

## 1. Introduction

The oxygen reduction reaction (ORR) is one of the most important electrochemical reactions in fuel cells [1] and metal–air batteries [2]. The ORR on Pt surfaces, such as for example, Pt single crystals [3,4], polycrystalline Pt electrodes [5], Pt nanoparticles (NPs) [6,7,8], Pt-based alloys [9,10], and core–shell Pt nanostructures [11,12], has been discussed in the last few decades because the use of Pt in fuel cell catalysts results in excellent catalytic performance. The knowledge of this catalytic reaction is valuable for developing cathode catalysts, and it can be applied to polymer electrolyte fuel cells (PEFCs) such as anion exchange membrane (AEM) and proton exchange membrane (PEM) fuel cells, which allow the fabrication of whole cells in alkaline and acid environments, respectively. A characteristic of AEMs is that hydroxyl ions produced by the ORR are transported from the cathode to the anode, in contrast to PEMs, where protons produced by the fuel oxidation reaction are transported from the anode to the cathode. AEMs have the following advantages over PEMs. (i) The fuel reaction in an alkaline environment is more active than that in an acid environment [13]. (ii) CO poisoning at the catalytic active sites is lower in an alkaline environment, because CO molecules that adsorb on the catalytic surface readily react with the abundant OH ions [13,14]. (iii) A non-noble catalyst can be used in an alkaline environment, in which the catalyst is protected against corrosion [15]. This is desirable not only for economic reasons, but also to conserve limited resources. Moreover, it has been reported that an AEM can eliminate the reaction of CO_2_ with the electrolyte, because there are no metal cations in the AEM [16,17]. As a result, the degradation of the electrolyte does not occur. For these reasons, AEMs have great potential to be utilized in direct methanol fuel cells (DMFCs) [16,17,18,19], direct hydrazine fuel cells (DHFCs) [15,20], and hybrid fuel cells [21,22]. However, since a reaction mechanism mainly based on the ORR on the Pt surface in an acidic solution may not be appropriate, it is necessary to understand the ORR mechanism in an alkaline solution to improve the performance and durability of AEM fuel cells consuming a liquid fuel such as methanol or hydrazine.

The ORR can be formulated on the basis of two representative reaction pathways: a “direct” four-electron reduction, and a “series” four-electron reduction [5,6,23]. In acid and alkaline media, these are represented as follows:

(i) The direct four-electron reduction in acid and alkaline media is represented by Equations (1) and (2), respectively.
(1)$$O_{2}+4H^{+}+4e^{-}\to 2H_{2}O$$
(2)$$O_{2}+2H_{2}O+4e^{-}\to 4{\mathrm{OH}}^{-}$$

(ii) The series pathway is shown in Equations (3)–(7). Oxygen is first reduced to a superoxide. Secondly, a peroxide species is formed as an intermediate (Equations (4) and (6) for acid and alkaline media, respectively), and this peroxide species reacts with a proton source (Equations (5) and (7) for acid and alkaline media, respectively).
(3)$$O_{2}+e^{-}⇌O_{2}{}^{-}$$
(4)$$O_{2}{}^{-}+2H^{+}+e^{-}⇌H_{2}O_{2}$$
(5)$$H_{2}O_{2}+2H^{+}+2e^{-}⇌2H_{2}O$$
(6)$$O_{2}{}^{-}+H_{2}O+e^{-}⇌{\mathrm{HO}}_{2}{}^{-}+{\mathrm{OH}}^{-}$$
(7)$${\mathrm{HO}}_{2}{}^{-}+H_{2}O+2e^{-}⇌3{\mathrm{OH}}^{-}$$

The ORR in an aprotic medium such as a high-pH solution proceeds via a different process from that in an acidic solution owing to the small number of protons [17]. Equations (3), (4), and (6) imply that the intermediate superoxide anion and peroxide species produced by the ORR are adsorbed on the catalyst surface. The ORR on noble metal surfaces in an alkaline medium has been investigated by performing electrochemical measurements with a rotating ring-disk electrode (RRDE). In addition, theoretical calculations based on density functional theory (DFT) predict the existence of several electrochemically adsorbed species, **O_2_, **O_2_^−^, **OOH, *OH, *O, and *H_2_O (where * and ** refer to the number of surface sites and chemisorbed species) [4,23,24,25]. Regarding the catalytic activity of Pt, it was reported that adsorbed OH oxidizes the Pt surface and deactivates the catalytic active sites [9], but it is uncertain how the Pt oxide is produced. Therefore, in order to elucidate the catalytic mechanism, it is necessary to clarify the electrochemically adsorbed species on a Pt surface in relation to key steps in the ORR pathway by direct observation under an in situ or in operando condition. Such knowledge will promote the development of an ideal cathode catalyst.

In general, the Pt 5*d* projected density of states (*d*-pDOS) moves to a lower energy upon chemical adsorption on the Pt surface [26]. Therefore, experimental observation of the electronic state reflecting the Pt *d*-pDOS under an in situ or in operando condition is important in order to clarify the process of electrochemical adsorption. Synchrotron radiation techniques allow the direct observation of the electronic state and atomic arrangement of a catalyst [6,7,8,10,11,12,20,27,28,29,30,31]. In particular, the use of hard X-rays makes it possible to carry out in situ and/or in operando experiments on the catalyst due to their high transmittance through gaseous, liquid, and solid materials. In situ observation of a Pt surface can be performed by X-ray absorption with the Pt L_3_ edge, which provides information on the Pt *d*-pDOS. In particular, high energy resolution fluorescence detection X-ray absorption spectroscopy (HERFD-XAS) is an appropriate spectroscopic method for studying the Pt 5*d* electronic state. Therefore, HERFD-XAS with an L_3_ edge has been utilized to examine the ORR on a Pt surface in an acidic solution and gas atmosphere [7,10,31], but has not yet been utilized for the reaction in an alkaline solution.

In our previous work, the Pt oxidation state in alkaline solution was discussed by the Cyclic Voltammetry with X-ray Absorption Fine Structure (CV-XAFS) method. However, it is uncertain what kinds of species are adsorbed on the Pt surface [30]. In the present work, electrochemical adsorbates on Pt NPs as typical cathode catalysts have been identified by using in situ HERFD-XAS in an alkaline solution. The identification was performed by a newly developed analysis method based on Δ*μ* X-ray absorption near-edge structure (XANES) measurement [6,8,12]. The ORR mechanism in an alkaline solution is discussed by comparing the experimental data obtained in an O_2_ atmosphere with those obtained in a N_2_ atmosphere as a function of the electrode potential.

## 2. Experiment

### 2.1. Electrochemical Cell

The electrochemical setup was similar to that in our previous study [30]. A Pt/C catalyst (TEC10E50E, Tanaka Kikinzoku Kogyo, Tokyo, Japan) with an average particle size between 2–4 nm [32,33] in diameter that was sprayed on carbon paper was placed on a carbon sheet as a working electrode and immersed in 1 M of KOH solution, which was saturated with N_2_ or O_2_. We assume that the ORR does occur in O_2_ atmosphere, but not in the N_2_ atmosphere. These gases were bubbled in the KOH solution over 1 h prior to the experiment, and maintained bubbling throughout the experiment. The electrolyte was circulated through an electrochemical cell by a pump (MP-2000, Tokyo Rikakikai, Tokyo, Japan). The reference electrode and current electrode were Hg/HgO (XR440, Radiometer) and a Pt coil, respectively. The potential at the working electrode was converted to that relative to a reversible hydrogen electrode (RHE) to remove the effect of the solution pH (we supposed that the value of pH is 14). The potential was controlled in the range from 0.04 V to 1.19 V versus RHE, which is in the potential window of the water, by using a potentiostat (model 611E, ALS Electrochemical Analyzer, BAS, Tokyo, Japan).

### 2.2. HERFD-XAS

In situ HERFD-XAS measurements were carried out at the BL11XU beamline in SPring-8 (Figure 1). The incident X-ray near the Pt L_3_ edge was monochromatized by double-crystal and channel-cut monochromators made of Si (111) and Si (400), respectively. The energy of the Lα_1_ fluorescence X-ray emitted from the catalyst was analyzed by Si (733) reflection. Both the incident and emission angles of X-rays were 45° from the sample surface. The fluorescence X-ray was collected using a two-dimensional detector (PILATUS 100K) located on the Rowland circle. XAS spectra were measured at each fixed potential about 600 s later after fluctuation of the current by having switched the potential had settled down.

### 2.3. Data Analysis

In general, an NP consists of surface and bulk atoms; thus, its XAS spectrum includes a metallic component originating from the bulk, which is not of interest here. Therefore, in order to obtain information on surface adsorption, it is necessary to subtract the information on the bulk metal from the XAS spectra of the metal NP catalyst. For this purpose, we adopted Δ*μ* XANES analysis, which is a method that is sensitive to surface adsorption [12]. The in situ HERFD-XAS intensity corresponds to an absorption coefficient (μ) that depends on the X-ray energy (*E*) and the electrochemical potential (*V*). μ consists of several components from the surface as well as from the bulk. This situation can be expressed as:(8)$$\mu \left(E,\;V\right)=\underset{i}{\sum }\alpha_{i}\left(V\right)\cdot \mu_{i}\left(E\right)$$

Here, $\alpha_{i}$ is the volume fraction of the *i*th component in the sample; thus:(9)$$\underset{i}{\sum }\alpha_{i}\left(V\right)=1$$

Δμ(E, V) is defined as the differential spectrum obtained from the Pt metal component as follows:(10)$$\Delta \mu \left(E,\;V\right)\equiv \mu \left(E,\;V\right)-\mu_{\mathrm{Pt}}\left(E\right)=\underset{i\neq \mathrm{Pt}}{\sum }\alpha_{i}\left(V\right)\cdot \mu_{i}\left(E\right)-\left(1-\alpha_{\mathrm{Pt}}\left(V\right)\right)\mu_{\mathrm{Pt}}\left(E\right)\;∵\underset{i}{\sum }\alpha_{i}\left(V\right)=\underset{i\neq \mathrm{Pt}}{\sum }\alpha_{i}\left(V\right)+\alpha_{\mathrm{Pt}}\left(V\right)=1$$

This calculation removes the bulk component from the spectrum at each *E* under potential *V* and emphasizes the *i*th surface component $\Delta \mu_{i}$:(11)$$\Delta \mu \left(E,\;V\right)=\underset{i\neq \mathrm{Pt}}{\sum }\alpha_{i}\left(V\right)\cdot \left\{\mu_{i}\left(E\right)-\mu_{\mathrm{Pt}}\left(E\right)\right\}\equiv \underset{i\neq \mathrm{Pt}}{\sum }\alpha_{i}\left(V\right)\cdot \Delta \mu_{i}\left(E\right)$$

In the case of Pt, although the Δ*μ* spectra with no contamination from the bulk metal were obtained by subtracting the spectrum of the metal-rich component at 0.38 V from the spectrum at each potential in our previous work [30], the spectrum observed at 0.34 V is close enough to that of the metal-rich component.
(12)$$\Delta \mu \left(E,\;V\right)\equiv \Delta \mu_{p}\left(E\right)=\mu_{p}\left(E\right)-\mu_{0.34V}\left(E\right)$$

XAS provides information on unoccupied orbitals because X-ray absorption is caused by the excitation of electrons from the core electronic state to the unoccupied state, which is the antibonding orbital in this case. In the other words, a higher intensity of XAS spectra will be observed at a lower energy if there are components of adsorption with weaker bonding to the metal surface. FEFF calculations result in the energy shift to the lower side in the *Δμ* spectra of Pt–O, Pt–OH, and Pt–OOH with this turn [8].

To analyze the electrode potential dependence of each component XAS, we introduced an idea of the rate of change of the Δ*μ* (RCD) intensity. RCD is defined as Δ*μ* normalized by the difference in Δ*μ* between two electrochemical potentials ($V_{ref}$, $V_{norm}$) as follows:(13)$$\mathrm{RCD}\left(E,\;V\right)\equiv \frac{\Delta \mu \left(E,\;V\right)-\Delta \mu \left(E,\;V_{ref}\right)}{\Delta \mu \left(E,\;V_{norm}\right)-\Delta \mu \left(E,\;V_{ref}\right)}=\frac{\sum_{i\neq \mathrm{Pt}}\Delta \mu_{i}\left(E\right)\left\{\alpha_{i}\left(V\right)-\alpha_{i}\left(V_{ref}\right)\right\}}{\sum_{i\neq \mathrm{Pt}}\Delta \mu_{i}\left(E\right)\left\{\alpha_{i}\left(V_{norm}\right)-\alpha_{i}\left(V_{ref}\right)\right\}}$$

The RCDs at $V_{ref}$ and at $V_{norm}$ are 0 and 1, respectively. In the case that the RCD is independent on the X-ray energy, there are only two components (Pt metal and the adsorbed species) between two electrochemical potentials.
(14)$$\mathrm{RCD}\left(E,\;V\right)=\frac{\Delta \mu_{ad}\left(E\right)\left\{\alpha_{ad}\left(V\right)-\alpha_{ad}\left(V_{ref}\right)\right\}}{\Delta \mu_{ad}\left(E\right)\left\{\alpha_{ad}\left(V_{norm}\right)-\alpha_{ad}\left(V_{ref}\right)\right\}}=\frac{\alpha_{ad}\left(V\right)-\alpha_{ad}\left(V_{ref}\right)}{\alpha_{ad}\left(V_{norm}\right)-\alpha_{ad}\left(V_{ref}\right)}$$

This equation indicates that there would be only one adsorbate around 0.34 V if RCDs calculated with different X-ray energies overlap each other. This indicates that RCD makes it easier to identify the adsorbate in various environments. The reason why RCD analysis can elucidate adsorbates is that the potential change causes an electron transfer in ORR.

In the case of a potential region being far from the metal-rich state, we assume that an oxidant (O_x_) is transposed to a reductant (R_x_) in the reaction process. This reaction is represented as:(15)$$\mathrm{Pt}-O_{x}+nH_{2}O+e^{-}⇌\mathrm{Pt}-R_{x}+{\mathrm{OH}}^{-}$$

The RCD is also written as:(16)$$\mathrm{RCD}\left(E,\;V\right)=\frac{\Delta \mu_{O_{x}}\left(E\right)\left\{\alpha_{O_{x}}\left(V\right)-\alpha_{O_{x}}\left(V_{ref}\right)\right\}+\Delta \mu_{R_{x}}\left(E\right)\left\{\alpha_{R_{x}}\left(V\right)-\alpha_{R_{x}}\left(V_{ref}\right)\right\}}{\Delta \mu_{O_{x}}\left(E\right)\left\{\alpha_{O_{x}}\left(V_{norm}\right)-\alpha_{O_{x}}\left(V_{ref}\right)\right\}+\Delta \mu_{R_{x}}\left(E\right)\left\{\alpha_{R_{x}}\left(V_{norm}\right)-\alpha_{R_{x}}\left(V_{ref}\right)\right\}}.$$

Under this assumption, the number of surface sites involved in the reaction is unchanged, and therefore, the sum of the component ratios (or concentrations of adsorbates) is constant.
(17)$$\underset{i\neq \mathrm{Pt}}{\sum }\alpha_{i}\left(V\right)=\alpha_{O_{x}}\left(V\right)+\alpha_{R_{x}}\left(V\right)\equiv c$$

The boundary condition for this reaction is:(18)$$\{\begin{array}{c} \alpha_{O_{x}}\left(V_{ref}\right)=0\\ \alpha_{R_{x}}\left(V_{ref}\right)=c \end{array}.$$

From these equations, the RCD intensity is calculated as:(19)$$\begin{array}{l} \mathrm{RCD}\left(E,\;V\right)=\frac{\Delta \mu_{O_{x}}\left(E\right)\alpha_{O_{x}}\left(V\right)+\Delta \mu_{R_{x}}\left(E\right)\left\{\alpha_{R_{x}}\left(V\right)-c\right\}}{\Delta \mu_{O_{x}}\left(E\right)\alpha_{O_{x}}\left(V_{norm}\right)+\Delta \mu_{R_{x}}\left(E\right)\left\{\alpha_{R_{x}}\left(V_{norm}\right)-c\right\}}\\ =\frac{\Delta \mu_{O_{x}}\left(E\right)\alpha_{O_{x}}\left(V\right)-\Delta \mu_{R_{x}}\left(E\right)\alpha_{O_{x}}\left(V\right)}{\Delta \mu_{O_{x}}\left(E\right)\alpha_{O_{x}}\left(V_{norm}\right)-\Delta \mu_{R_{x}}\left(E\right)\alpha_{O_{x}}\left(V_{norm}\right)}=\frac{\alpha_{O_{x}}\left(V\right)}{\alpha_{O_{x}}\left(V_{norm}\right)}. \end{array}$$

This equation indicates that the RCD curves with different X-ray energies coincide with each other as a function of the potential if the oxidant is replaced by the reductant.

## 3. Results and Discussion

### 3.1. Δμ Analysis

Figure 2a,b show the XAS spectra of the Pt catalyst in KOH saturated with N_2_ and O_2_, respectively. The spectra were normalized with the intensity around 11,573 eV, which is near the isosbestic point reported previously [11,12]. It was observed that the shoulder peak in XAS spectra appears with a change of the electrochemical potential in both atmospheres. To enhance this information on Pt oxidation, the Δμ spectra in N_2_ and O_2_ atmospheres were obtained by Equation (12) (Figure 3 and Figure 4, respectively). As shown in Figure 3, a negative peak at 11,566.75 eV was observed below 0.34 V, a broad peak at 11,567.50 eV was observed from 0.54 V to 0.94 V, and a sharp peak at 11,569.75 eV was observed from 0.74 V to 1.19 V in N_2_. As shown in Figure 4, a negative peak at 11,566.75 eV was observed below 0.34 V, a sharp positive peak at the same energy was observed from 0.44 V to 1.04 V, a positive peak at 11,569.75 eV was observed from 0.84 V to 1.19 V, and a negative sharp peak at 11,566.50 eV was observed at the two highest potentials (1.14 V and 1.19 V) in O_2_. From these observations, the change of the Δ*μ* spectra in the high potential side was found to start from around 0.74 V. This means that more than two adsorbate components should be considered above 0.74 V to discuss the adsorbed ORR intermediates. This indicates that adsorbates can be investigated simply by considering RCDs calculated with Equation (14) below 0.74 V. Therefore, the potential pairs of ($V_{ref}$, $V_{norm}$) are (0.04 V, 0.34 V) and (0.34 V, 0.74 V). Moreover, the energies of 11,566.75 eV, 11,568.00 eV, and 11,569.75 eV were selected because of the following reasons: 11,566.75 eV is the energy of the negative peak observed below 0.34 V; 11,568.00 eV is near the energy of the peak observed from 0.34 V to 0.74 V, and is not significantly affected by the peak at 11,566.75 eV, but is affected by the peak at 11,567.50 eV; and lastly, 11,569.75 eV is the energy of the peak observed above 0.74 V. The dashed RCD curves for 11,566.00 eV, 11,567.50 eV, 11,569.00 eV, and 11,572.00 eV have been added in RCDs to make the results of adsorptions (discussion about adsorptions) clearer.

### 3.2. Adsorbates on Pt Surface at Low Potential (from 0.04 V to 0.74 V) in N_2_ Atmosphere

In the N_2_ atmosphere, the typical RCD curve (Figure 5) for 11,566.75 eV in Figure 3 shows the electrochemical potential dependence of the negative peak below 0.34 V observed in Figure 2. It can be seen that the RCD curves that were analyzed using Equation (13)—with incident X-ray energies from 11,566.00 eV to 11,568.00 eV, which form part of the negative peak—lie on a single line. This means that the RCD is independent of the X-ray energy at potentials below 0.34 V. Since this negative peak is known to be a result of hydrogen underpotential deposition [3,7], it can be concluded that only hydrogen is adsorbed on the Pt surface in the potential range from 0.04 V to 0.34 V. This reaction is written as:(20)$$\mathrm{Pt}+H_{2}O+e^{-}⇌\mathrm{Pt}-H+{\mathrm{OH}}^{-}$$

Above 0.34 V, hydrogen completely desorbs from platinum, and this region is known as the “butterfly region” owing to the highly reversible adsorption of hydroxyls [5].
(21)$$\mathrm{Pt}-\mathrm{OH}+e^{-}⇌\mathrm{Pt}+{\mathrm{OH}}^{-}$$

In the N_2_ atmosphere, other adsorbates on the Pt surface are unlikely at low potential because the alkaline electrolyte without the oxygen molecule suppresses the evolution of other species. For example, it is difficult to form peroxide (OOH) species and water molecules, which need three hydroxyl species.
(22)$$\mathrm{Pt}-\mathrm{OOH}+H_{2}O+3e^{-}\leftarrow \mathrm{Pt}+3{\mathrm{OH}}^{-}$$

In Figure 5, the RCD curves almost overlap each other above 0.34 V, indicating that hydroxyl adsorption mostly occurs. However, the RCD intensity at 0.64 V obtained from the curve for 11,566.00 eV does not agree with the values obtained with other X-ray energies. This difference is considered to be caused by the existence of a different component, such as hydroxyl adsorbed at other sites of the Pt surface. It is known that different redox potentials exist, corresponding to adsorption on different crystal planes, as observed by cyclic voltammetry [3]. In the present study, the three RCD curves (excluding the curve for 11,566.00 eV) almost coincide with each other, and the X-ray energies with which these three curves are calculated are higher than 11,566.00 eV. Therefore, the above result suggests the existence of a small amount of weakly bonded hydroxyls adsorbed on a different surface site from that where hydroxyl adsorption mainly occurs.

### 3.3. Adsorbates on Pt Surface at Low Potential (from 0.04 V to 0.74 V) in O_2_ Atmosphere

In the O_2_ atmosphere (Figure 4), a negative peak originating from hydrogen adsorption was also observed. However, the energy of the positive peak above 0.34 V is different from that for hydroxyl adsorption observed in N_2_ atmosphere. This implies that there is at least one more adsorption species other than the hydroxyl group. This energy is 11,566.75 eV, which is almost the same as that for hydrogen adsorption and lower than the peak energy for hydroxyl adsorption. This means that the bonding of this new adsorbate to the Pt surface is weaker than that of the hydroxyl adsorbate. Also, by considering the results in the N_2_ atmosphere, it was reasonably concluded that hydrogen adsorbates cannot exist above 0.34 V. Therefore, we expect that this new adsorbate could be a superoxide anion [5,6,23,24,34] formed similarly to Equation (3):(23)$$x\mathrm{Pt}+O_{2}+e^{-}⇌{\mathrm{Pt}}_{x}-O_{2}{}^{-}$$
where *x* is 1 or 2 for the adsorption model on the Pt surface, where the adsorbate has the end-on or side-on configuration, respectively. Another candidate is a peroxide, but it was reported that the Pt electrode was adsorbed by the superoxide anion, but not the peroxide in alkaline medium (pH = 11), as observed by surface-enhanced infrared reflection absorption spectroscopy [23]. Therefore, from these considerations, it is reasonable to conclude that the adsorbate observed in the present experiment is a superoxide anion. A superoxide anion was observed in an aprotic medium in a previous study because of its long lifetime [5]. To investigate other components, we also analyzed the RCD intensity in the O_2_ atmosphere (Figure 6). As shown in Figure 6, the curves roughly coincide with each other in the potential range between 0.34–0.74 V. This finding suggests less hydroxyl adsorption on the surface in the O_2_ atmosphere. The adsorption of the superoxide anion on the Pt surface is the first step in the ORR, and the observation of the superoxide anion but not the hydroxyl group is interpreted by considering the rate-determining step of the ORR as follows. According to Equations (24) and (25), the reaction frequency of the superoxide anion is decreased by the low H_2_O concentration because of the high hydroxyl concentration in the alkaline solution [5].
(24)$${\mathrm{Pt}}_{x}-O_{2}{}^{-}+H_{2}O⇌{\mathrm{Pt}}_{x}-\mathrm{OOH}+{\mathrm{OH}}^{-}$$
(25)$${\mathrm{Pt}}_{x}-\mathrm{OOH}+H_{2}O+2e^{-}\to {\mathrm{Pt}}_{x-1}+\mathrm{Pt}-\mathrm{OH}+2{\mathrm{OH}}^{-}$$

The peroxide and hydroxyl species were not observed in the experiment, although the reaction in Equation (24) occurs independently of the potential. This can be explained by an assumption that the reactions in Equations (25) and (21) are fast in the O_2_ atmosphere.

As mentioned in Section 3.2, only hydrogen adsorbs on the Pt surface below 0.34 V in the N_2_ atmosphere. However, the RCD intensity below 0.34 V is scattered for the sample in the O_2_ atmosphere, as shown in Figure 6. This suggests that the adsorbates are not only hydrogen but also another species, by having been explained in Equations (13) and (14). The Pt surface in the O_2_ atmosphere is more oxidized than that in the N_2_ atmosphere, because the adsorption of the superoxide anion already exists at 0.34 V. This prediction is supported by the Δ*μ* spectrum at 0.34 V obtained in the O_2_ atmosphere (Appendix A). Such oxidative adsorption was observed in our previous work [30] in which oxygen rapidly reacted or interfered with adsorbed hydrogen. In other words, the superoxide anion, which is attributed to oxygen adsorption on the Pt surface, reacts with the hydrogen from water, which acts as a proton source (Equations (24) and (25)), and therefore the amount of observed hydrogen adsorption was small.

### 3.4. Pt Oxidation at High Potential (above 0.74 V) in N_2_ Atmosphere

In the high-potential region, a large peak was observed at around 11,569.75 eV for the sample in the N_2_ atmosphere (Figure 2). The RCD curves for 11,569.00 eV, 11,569.75 eV, and 11,572.00 eV, which form part of this peak, have been added to Figure 7. However, the data are scattered with energy between 11,566.75–11,568.00 eV in Figure 7. This suggests that there are three or more components in the high-potential region, as indicated by Equations (13) and (19). The curves corresponding to energies over 11,569 eV indicate the existence of adsorbed species other than hydrogen, the hydroxyl group, and the superoxide anion. This is interpreted as being due to the adsorption of atomic oxygen by the following reaction:(26)$$\mathrm{Pt}-O+H_{2}O+e^{-}⇌\mathrm{Pt}-\mathrm{OH}+{\mathrm{OH}}^{-}$$

The adsorbate of atomic oxygen reacts with water and becomes hydroxyl adsorption above 0.74 V in ORR. This reaction is the same as that in Equation (15). In fact, the RCD curves normalized between 0.74–0.94 V almost coincide with each other (Appendix A). However, since other reactions occur above 0.94 V, the curves do not perfectly overlap. The scattered data of the RCD curves can be understood by the existence of Pt oxides, which is produced by the following reaction process.
(27)$${\mathrm{PtO}}_{2}+H_{2}O+2e^{-}⇌\mathrm{Pt}-O+2{\mathrm{OH}}^{-}$$

The existence of oxides is deduced from the curve for 11,566.75 eV (filled square), which abruptly changes above 0.94 V (Figure 7). This shows the potential, which starts the construction of Pt oxide, but the Δ*μ* spectrum of this oxide is different from that of the reference sample (Appendix A). The peak energy of the Δ*μ* spectrum at 1.19 V for the sample in the N_2_ atmosphere is lower than that of the reference sample of Pt oxide, whose structure is β-PtO_2_. This Pt oxide has weaker bonding than β-PtO_2_, as discussed in Section 3.1. In addition, it is known that the Pt oxidation state progresses from α-PtO_2_ to β-PtO_2_ over its potential [27,28,29]. Therefore, we expect that this oxide could be assigned as α-PtO_2_. By considering these findings, it can be concluded that the observed data show the evolution of α-PtO_2_ (Pt oxidation). However, the progress of “place exchange” (mentioned below) is not enough to reach final form of α-PtO_2_, and this indicates that Pt–O is mainly the adsorption of the oxygen in the N_2_ atmosphere.

### 3.5. Pt Oxidation at High Potential (above 0.74 V) in the O_2_ Atmosphere

As it can be seen in Figure 8, the RCD curves do not coincide with the curves for the X-ray energies from 11,566.75 eV to 11,572.00 eV. This indicates the existence of at least three components. In the O_2_ atmosphere, the superoxide anion is adsorbed above 0.34 V, and the adsorption of atomic oxygen takes place above 0.74 V, as can be understood by considering the results obtained for the sample in the N_2_ atmosphere. This reaction can be written as:(28)$${\mathrm{Pt}}_{x}-O_{2}{}^{-}+H_{2}O+e^{-}⇌\mathrm{Pt}-O+{\mathrm{Pt}}_{x-1}+2{\mathrm{OH}}^{-}$$

If *x* is equal to 1 in the case that the oxidant is the superoxide anion and the reductant is the atomic oxygen in Equation (15), then the RCD curves will coincide with each other. However, the data do not indicate such a case, as shown in Appendix A, between 0.74–0.94 V, where the RCD curves observed in the N_2_ atmosphere lie on a single line (Appendix A). If x is equal to 2, the superoxide anion on the two platinum atoms becomes the adsorbate of atomic oxygen, and no hydroxyl species are observed. In this case, the RCD curves should not coincide. Since the experimental data in Appendix A do not coincide with the curves between 0.74–0.94 V, x is concluded to be 2; that is, the superoxide anion adsorbs with the side-on configuration.

In Figure 7, an abrupt change in the curve comparing filled squares is observed above 0.94 V. As discussed in Section 3.4 in the case of the N_2_ atmosphere, this change is caused by the formation of α-PtO_2._ However, the dramatic reduction in intensity at 11,566.75 eV is observed above 1.14 V, as shown in Figure 3. This feature was previously described as a “place exchange” involving the penetration of oxygen into the Pt bulk, as reported by Sasaki et al. [12], and α-PtO_2_ is mainly formed in this potential region. ORR starting below 1.02 V [30] does not occur when α-PtO_2_ starts to form.

### 3.6. ORR Mechanism in Alkaline Solution

Here, we describe the overall ORR mechanism by taking into account the present analysis. The first step of the ORR is the adsorption of the superoxide anion with the side-on configuration (Equation (23)). The second step is the dissociation or non-dissociation process. When the dissociation of the superoxide anion occurs (Equation (28)), we can obtain a potential of more than 0.74 V, as discussed in Section 3.5. The atomic oxygen that evolves from this reaction reacts with water (Equation (26)). Since no or few hydroxyl species were observed in the O_2_ atmosphere, this reaction process is the rate-determining step in the ORR. This implies that the cell performance can be improved by changing the bond strength of the atomic oxygen adsorbed on the platinum surface. When no dissociation of the superoxide anion occurs (Equation (24)), peroxide species can be evolved, and we obtain a potential lower than 0.74 V (Equation (25)). In this case, this reaction process (Equation (24)) is the rate-determining step in the ORR. This step should be improved by using a lower pH and changing the stability of the superoxide anion adsorbed on the platinum surface. Finally, hydroxyl species desorb from the platinum surface in both cases (Equation (16)). The protic adsorption from water (Equation (20)) occurs at a low potential, but hydrogen adsorption cannot occur stably in an O_2_ atmosphere because the hydrogen reacts with the superoxide anion in the elemental reactions (Equations (24) and (25)).

## 4. Conclusions

To identify the ORR mechanism in an alkaline solution, we discussed adsorbates on a platinum surface and its oxidation in N_2_ and O_2_ atmospheres as a function of the electrode potential in the alkaline solution by analyzing Δ*μ* XANES spectra with such high accuracy measured by the in situ HERFD-XAS technique. For this analysis, we introduced the newly developed RCD method, which can analyze the number of components making up the Δ*μ* spectrum. By comparing the RCD curves calculated from the Δ*μ* spectra, various components, such as **O_2_^−^, *OH, *O, *H, and α-PtO_2_, can be distinguished, which are adsorbed and form an oxide on the platinum surface. Using the RCD method, it was analyzed that the superoxide anion adsorbs on two platinum atoms with a side-on configuration. From the present results, we conclude that in order to increase the catalytic activity in an alkaline solution, the stability of atomic oxygen and the low stability of superoxide anion/peroxide adsorption on the platinum surface should be enhanced. This can be achieved by lowering the pH of the alkaline electrolyte.

## Figures and Tables

**Figure 1 nanomaterials-09-00642-f001:**
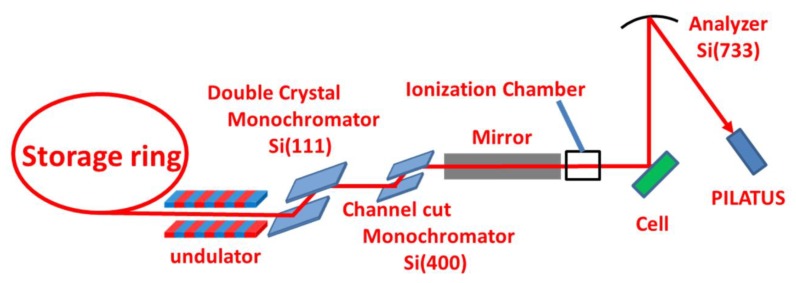
The optics of high energy resolution fluorescence detection X-ray absorption spectroscopy (HERFD-XAS) experiment at the BL11XU beamline in SPring-8.

**Figure 2 nanomaterials-09-00642-f002:**
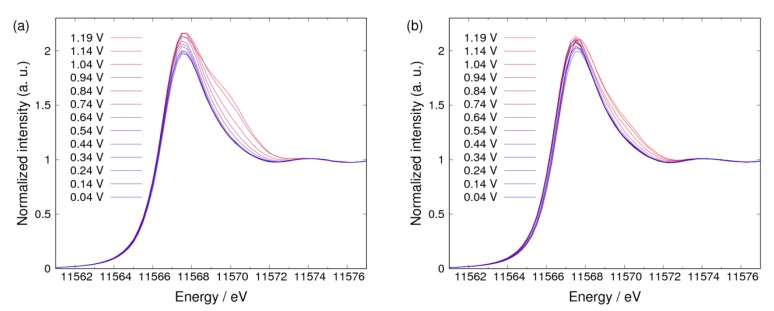
Pt L_3_ HERFD-XAS spectra of Pt nanoparticles (NPs) under potential control in 1 M of KOH saturated with (**a**) N_2_ and (**b**) O_2_.

**Figure 3 nanomaterials-09-00642-f003:**
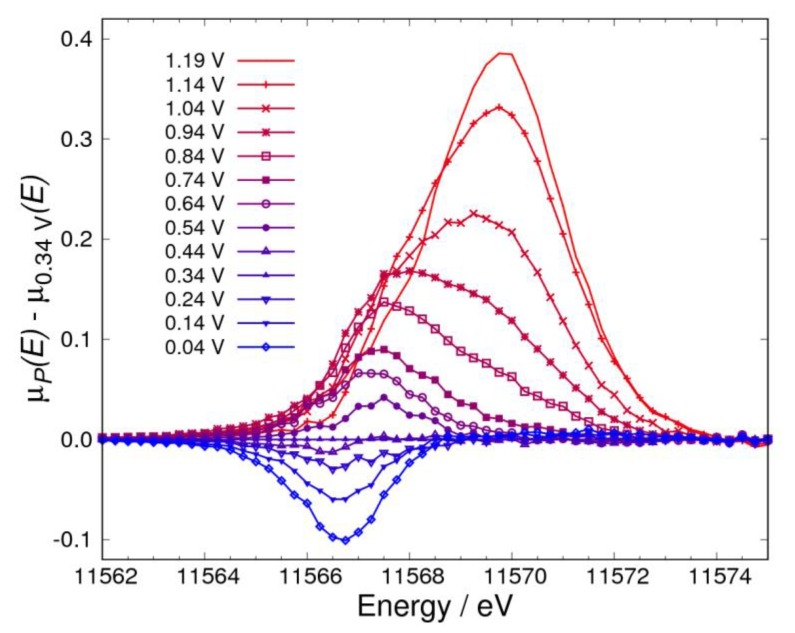
Δ*μ* XANES spectra for the catalyst calculated by $\mu_{p}\left(E\right)-\mu_{0.34V}\left(E\right)$ under potential control in 1 M of KOH saturated with N_2_.

**Figure 4 nanomaterials-09-00642-f004:**
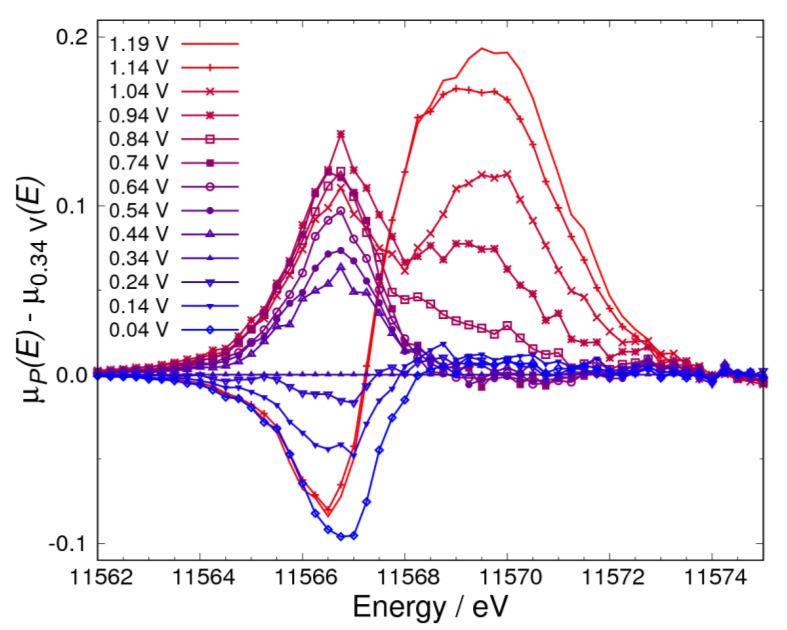
Δ*μ* XANES spectra for the catalyst calculated by $\mu_{p}\left(E\right)-\mu_{0.34V}\left(E\right)$ under potential control in 1 M of KOH saturated with O_2_.

**Figure 5 nanomaterials-09-00642-f005:**
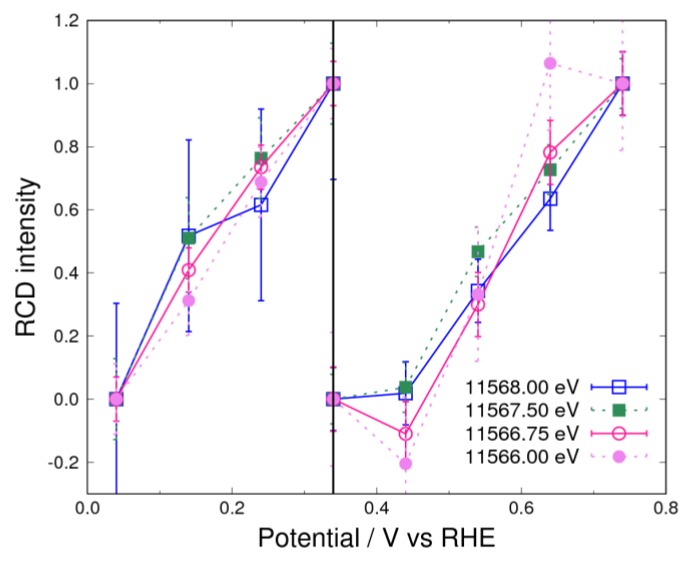
Rate of change of the Δ*μ* (RCD) curves for the controlled potential below 0.74 V in 1 M of KOH saturated with N_2_. The standard potentials are 0.04 V, 0.34 V, and 0.74 V. The circle and square curves represent the negative and positive peak of 11,566.75 eV and 11,567.5 eV, respectively. The dashed lines complement the explanation for RCD.

**Figure 6 nanomaterials-09-00642-f006:**
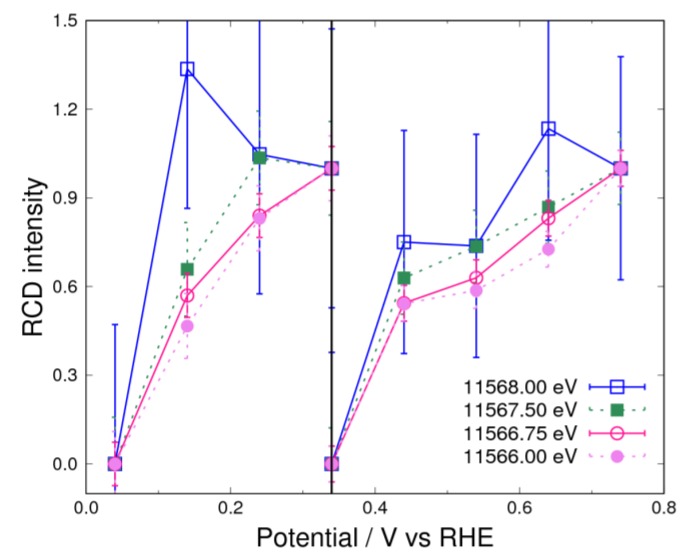
RCD curves for the controlled potential below 0.74 V in 1 M of KOH saturated with O_2_. The standard potentials are 0.04 V, 0.34 V, and 0.74 V. The circle and square curves represent the negative and positive peak of 11,566.75 eV and 11,567.5 eV, respectively. The dashed lines complement the explanation for RCD.

**Figure 7 nanomaterials-09-00642-f007:**
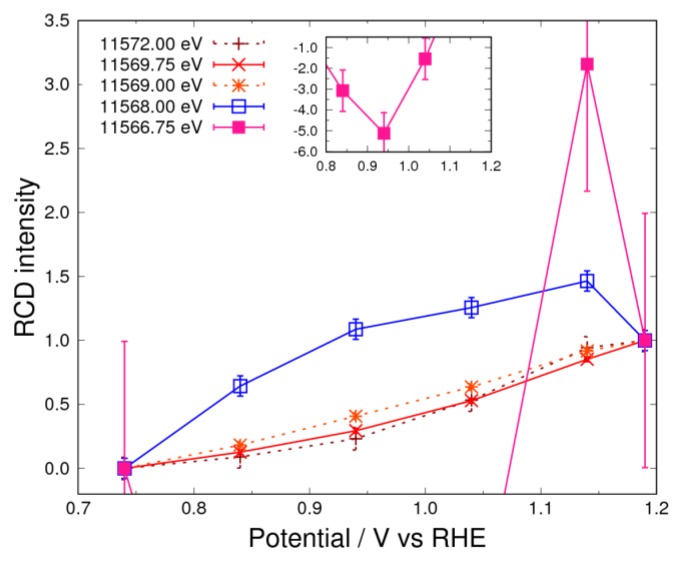
RCD curves for the controlled potential above 0.74 V in 1 M of KOH saturated with N_2_. The standard potentials are 0.74 V and 1.19 V. The filled and open square curves represent the negative and positive peak of 11,566.75 eV and 11,568.00 eV, respectively. The dashed lines complement the explanation for RCD.

**Figure 8 nanomaterials-09-00642-f008:**
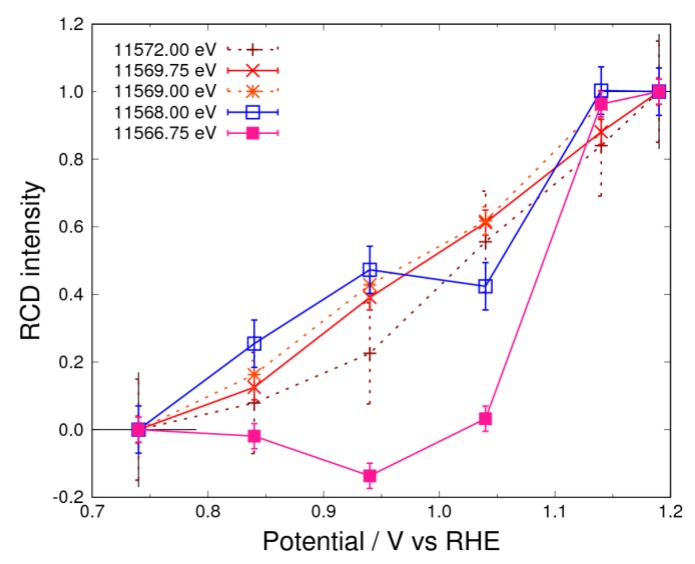
RCD curves for the controlled potential above 0.74 V in 1 M of KOH saturated with O_2_. The standard potentials are 0.74 V and 1.19 V. The filled and open square curves represent the negative and positive peak of 11,566.75 eV and 11,568.00 eV, respectively. The dashed lines complement the explanation for RCD.

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
