# Peer review of "Electrochemical Adsorption on Pt Nanoparticles in Alkaline Solution Observed Using In Situ High Energy Resolution X-ray Absorption Spectroscopy"

_nanomaterials, 2019, doi:10.3390/nano9040642_

Round 1
Reviewer 1 Report
This is a very good manuscript that is easy to read, follow, and understand. The authors are encouraged to add more background references in the introduction as the literature surveyed is a bit sparse. The authors are also recommended to change the color scheme in Figures 5,6 to remove/replace the light blue which is difficult to see on the screen. The manuscript should be considered acceptable for publication once the authors address these minor concerns and go through further proofreading for typos, etc.
Author Response
Response: We thank the reviewer 1 for his/her careful reading of our manuscript and commenting the important issues. We modified English in our manuscript, and we replace the light blue in Figure 5, 6 to dark green.

Reviewer 2 Report
The English need to be improved in the paper, it happens that the intend behind the sentences are compromised due to poor gramma. Also, the fact that abbreviations are not always explained (e.g. what does RCD stand for?) really makes the text hard to follow. Furthermore, I would request the authors to be more mindful when to refer to which figures. NB It seems that to formats of citations have been used.
My main concerns are the following:
1) The method proposed though surface sensitive does not give spectroscopic information about what is on (or coming of the Pt surface). Rather you assign changes to hydroxyl, superoxide, hydrogen etc. But how do you know? E.g. there’s plenty of information suggesting cations (such as K) can adsorb on Pt at low potential. I.e. maybe changes in signal is not just adsorption/desorption of single species but co-adsorption/co-desorption of one or more species. I think this is important to touch upon.
2) Your analysis relies on a set assumption (eq. (16)) that the surface concentration of adsorbates are constant. How good is this assumption and is that correct understood? If correct how would you address that theory conjoined with experiments on Pt(111) suggest Pt the saturation coverages are potential dependent (charge isotherms; see work by Feliu or Rossmeisl)? As I understand it one would have to weigh with coverages of the species to be able to say anything in terms of peak changes.
3) When conducting the experiments, it appears you went from very low potential to high potential (correct?) If so did you try to reverse the measuring orders of the potentials to see whether it was reversible. It is well-known that Pt in alkaline media is i) prone to adsorb metal hydride impurities in a quasi-stable configuration on the surface (which would lead to changes in you relative XAS coefficient). ii) In alkaline media RHE electrodes are not as stable as in acid (especially if experiements take more than 1 h), hence to exact the potential one is required to saturate with H2 and detect the OC potential. How would you address this and how would such considerations impact your conclusions?
4) Earlier work by Magnussen and co-workers employed SAXS to detect surface roughening of Pt(111) in acid at 1.17 V vs. RHE. It would not be completely unexpected that similar roughening could happen in alkaline even at lower potential as roughening from oxide formation should be more prevalent. Couldn’t surface roughening and thus changes in surface coordination of Pt account for some of your observed changes in adsorption coefficient?
5) You suggest that ORR activity of Pt is improved in alkaline media; this is categorically wrong. Pt NPs in alkaline media presents with roughly half the activity compared to when in HClO4, see work by Arenz and Nesselberger). Exactly what do you intend with the sentence in lines 21-23? In Line 68 you state that selfposioning of OH on the Pt (through oxidation) limits activity, hence in alkaline this effects should be more predominant, right?
6) I assume that in line 100-101 you mean to say that “ORR does not occur when not in a electrochemically connected to the electrochemical cell” or similar, right?
7) Line 159-160 you conclude on H adsorb in the low potential region in 1 M KOH, what is the proof or reference for this?
8) You specifically state that the XAS spectrum of the earlier spectrum at this potential “is close enough” to you herein presented data. The data you refer to is also in 1 M KOH so im a little bit confused by what you are conveying to me and its implications. To me it basically reads we have to almost similar measurement but for one Pt measurements there may or may not be something slightly changes the spectrum e.g. contaminants etc. Also the KOH you used what purity/type was it?
9) I’m concerned about your conclusions; you basically assign changes in adsorption to specific species without any proof that it is these and not just more/less hydroxyl adsorbing on the surface or changes in surface coverages from Pt dissolution (and consequent increased under coordination form roughening) or contamination form metallic impurities.
Author Response
The English need to be improved in the paper, it happens that the intend behind the sentences are compromised due to poor gramma. Also, the fact that abbreviations are not always explained (e.g. what does RCD stand for?) really makes the text hard to follow. Furthermore, I would request the authors to be more mindful when to refer to which figures. NB It seems that to formats of citations have been used.
Response: We thank the reviewer 2 for his/her careful reading of our manuscript and commenting the important issues. We respond the reviewer’s concerns below.
We revised the English in red in the manuscript. We hope that these are more understandable than the previous manuscript.
Abbreviation of RCD is written in line 18, and 147.
My main concerns are the following:
1) The method proposed though surface sensitive does not give spectroscopic information about what is on (or coming of the Pt surface). Rather you assign changes to hydroxyl, superoxide, hydrogen etc. But how do you know? E.g. there’s plenty of information suggesting cations (such as K) can adsorb on Pt at low potential. I.e. maybe changes in signal is not just adsorption/desorption of single species but co-adsorption/co-desorption of one or more species. I think this is important to touch upon.
The allocation of adsorbed species is based on the report of past electrochemical measurement and the report of simulation of XAS (FEFF) in reference 8.
The cation in the aqueous solution is only a potassium, and by considering the ionization tendency of potassium, the probability of electrodeposition of potassium in the aqueous solution must be low. Then, even if potassium is adsorbed on the Pt surface, a character of the adsorption could be a physisorption, not a chemisorption.
If co-adsorption (more than two kinds of chemisorption) exist, two or more different electronic states from that of a metal Pt surface should be formed. The RCD analysis newly developed in the present work can distinguish between the adsorption of single species and more than two kinds of species. In this case, the RCD curves should not lie on a single line with several incident X-ray energies, because the spectra reflect the mixture of two or more species. But, as you can see in the left part of Fig. 5, the RCD is independent on the X-ray energy at potentials below 0.34 V. This means that even if co-adsorption occurs, it should not be chemical adsorption that changes the electronic state.
2) Your analysis relies on a set assumption (eq. (16)) that the surface concentration of adsorbates are constant. How good is this assumption and is that correct understood? If correct how would you address that theory conjoined with experiments on Pt(111) suggest Pt the saturation coverages are potential dependent (charge isotherms; see work by Feliu or Rossmeisl)? As I understand it one would have to weigh with coverages of the species to be able to say anything in terms of peak changes.
Since the RCD curve is normalized between two potentials, it corresponds to the change in coverage (the concentration of Pt decreases as the adsorption of something increases). On the high potential side, if the concentration of any one of Pt, Pt-Ox, and Pt-Rx is constant, eq. (16) can be rewritten to eq. (19). Otherwise, the RCD curves do not match as shown in equation 19 because the coverage changes individually.
3) When conducting the experiments, it appears you went from very low potential to high potential (correct?) If so did you try to reverse the measuring orders of the potentials to see whether it was reversible. It is well-known that Pt in alkaline media is i) prone to adsorb metal hydride impurities in a quasi-stable configuration on the surface (which would lead to changes in you relative XAS coefficient). ii) In alkaline media RHE electrodes are not as stable as in acid (especially if experiements take more than 1 h), hence to exact the potential one is required to saturate with H2 and detect the OC potential. How would you address this and how would such considerations impact your conclusions?
The experiment was conducted by scanning the potential from high to low side, but we have not tested reverse order. Therefore, we have not checked whether or not the experimental result is reversible. However, initialization of the surface state was done by performing CV measurement just before the XAS experiment at each potential. Then it can be considered as a nearly reversible state.
The experiment uses Hg / HgO electrode (1 M KOH) instead of RHE electrode. We calculated RHE with using the potential value of Hg / HgO electrode. We do not measure the true RHE potential.
4) Earlier work by Magnussen and co-workers employed SAXS to detect surface roughening of Pt(111) in acid at 1.17 V vs. RHE. It would not be completely unexpected that similar roughening could happen in alkaline even at lower potential as roughening from oxide formation should be more prevalent. Couldn’t surface roughening and thus changes in surface coordination of Pt account for some of your observed changes in adsorption coefficient?
Since the Pt sample we studied is a nanoparticle shape with an average size of 2-4 nm in diameter, the results are averaged about the surface direction. In other words, the results must be insensitive to the morphology of the sample even if roughening happens. And also, since the XAS measurement was done by the observation of the Lα1-fluorescence X-ray emitted from the sample, the result is insensitive to the morphology of the sample.
5) You suggest that ORR activity of Pt is improved in alkaline media; this is categorically wrong. Pt NPs in alkaline media presents with roughly half the activity compared to when in HClO4, see work by Arenz and Nesselberger). Exactly what do you intend with the sentence in lines 21-23? In Line 68 you state that selfposioning of OH on the Pt (through oxidation) limits activity, hence in alkaline this effects should be more predominant, right?
We did not intend saying that the ORR activity of the alkaline media is higher than that of the acidic media. But we would like to discuss how the ORR activity becomes higher in the alkaline media. Our conclusion in the present results is that the stability of atomic oxygen and the low stability of superoxide anion/peroxide adsorption on the platinum surface should be enhanced, which is mentioned at the conclusion chapter in the manuscript.
6) I assume that in line 100-101 you mean to say that “ORR does not occur when not in a electrochemically connected to the electrochemical cell” or similar, right?
No, it is nothing related to ORR whether or not the system is connected to the electrochemical cell. What we would like to say in this sentence is that ORR does not occur if there is no oxygen (only nitrogen) in the solution. We revise the sentence as follows:
“We assume that the ORR does occur in O2 atmosphere, but not in the N2 atmosphere.”
7) Line 159-160 you conclude on H adsorb in the low potential region in 1 M KOH, what is the proof or reference for this?
According to the reference 3, 7 in the manuscript, it is reported that an underpotential deposition of hydrogen occurs near the hydrogen evolution potential.
8) You specifically state that the XAS spectrum of the earlier spectrum at this potential “is close enough” to you herein presented data. The data you refer to is also in 1 M KOH so im a little bit confused by what you are conveying to me and its implications. To me it basically reads we have to almost similar measurement but for one Pt measurements there may or may not be something slightly changes the spectrum e.g. contaminants etc. Also the KOH you used what purity/type was it?
In our previous CV-XAFS study, we observed that Pt is metallic state at 0.38 V and is also metallic state too at 0.34 V. There is no other metal impurity in KOH.
9) I’m concerned about your conclusions; you basically assign changes in adsorption to specific species without any proof that it is these and not just more/less hydroxyl adsorbing on the surface or changes in surface coverages from Pt dissolution (and consequent increased under coordination form roughening) or contamination form metallic impurities.
As the reviewer mentioned, we did not observe directly the adsorption species. However, the conclusion regarding the adsorption species is drawn based on the information of XAS, the published result of FEFF calculation and the change of adsorbed species depending on the potential discussed in the past papers. These results can be reasonably explained without any contradiction each other.
It is possible to adsorb other species than those we considered. We think that these may not be detectable because they are covered in the spectrum from those of the major adsorbed species.

Reviewer 3 Report
The authors used XANES spectra and newly developed RCD method to study the ORR mechanism in an alkaline soultion. The adsorbed species and their reactions on the Pt surface can be resolved. The ORR behaviors in an alkaline solution as a function of the electrode potential are compared in N2 and O2 atmosphere, respectively. The result is significant and well discussed. I suggest the manuscript to be published as is.
Author Response
Response: We thank the reviewer 3 for his/her careful reading of our manuscript and commenting the important issues. We modified English in our manuscript.

Round 2
Reviewer 2 Report
I would urge the authors to go through the grammar once more e.g. in the abstract it says "...rate of change of the Δμ (RCD) which is an analysis method..." where the "an" have been included by reviewer. Also some places N2 is written as N2 and captialization is used certain places by mistake.
Content-wise I believe the experiments are interesting but I disagree with quite a few aspects of their interpretations. However, this may still be of interest to the field. My main concern is the superoxide detection. Authors reference to the nice paper of Adzic et al [23], this suggests superoxide at the surface of Pt in alkaline from the range from ca 0 to 0.7 V vs RHE. What I found curious is that superoxide was observed below 0.34 V by the authors, below 0.34 V there is nothing but H. Now the point is the authors have assigned some signals from the XAS to the pressence of different species, but the assignage is determined by a model (FEFF) and a fixed background, but given PtO2 and metalic Pt XAS differs vastly in backround and the fact that small Pt Nanoparticle makes the system exceptional sensitive to surface changes e.g. from coordination and and oxidation I believe the the authors to a higher degree should make it clear that the species assignmet is a working hypothesis.
Author Response
Response to Reviewer 2 Comments
Response: We thank the reviewer 2 for his/her careful reading of our manuscript and commenting the important issues. We respond the reviewer’s concerns below.
(1) I would urge the authors to go through the grammar once more e.g. in the abstract it says "...rate of change of the Δμ (RCD) which is an analysis method..." where the "an" have been included by reviewer. Also some places N2 is written as N2 and captialization is used certain places by mistake.
We revised the English in red in the manuscript. We hope that these are more understandable than the previous manuscript.
Line 37: “in” -à erase
Line 73: “condition Such knowledge….”à “condition. Such knowledge….”
Line 84: “the 5d electronic state of platinum”à “the Pt 5d electronic state”
Line 97: “in our previous study.31”-à “in our previous study [30].”
Line 179-180: “This fact indicates that adsorbates were investigated by considering RCDs below 0.74 V in order to discuss eq. (14).”-à ” This fact indicates that adsorbates can be investigated by considering RCDs calculated with eq. (14) below 0.74 V.”
Line 237: “in the N2 atmosphere,”-à “in the N2 atmosphere,”
Line 241: “Another candidate is peroxide,”-à “Another candidate is a peroxide,”
Line 242: “adsorbed by superoxide anion but not peroxide”-à “adsorbed by the superoxide anion but not the peroxide”
Line 244: “observed in present experiment is superoxide anion.”-à “observed in the present experiment is a superoxide anion.
Line 281: “by eq. 13 and eq. 19.”-à “by eq. (13) and eq. (19).”
Line 282: “energies of over 11569 eV”-à” energies over 11569 eV”
Line 293: “as eq. 15.”-à “as eq. (15).
Line 323-324: “reductant is atomic oxygen”-à “reductant is the atomic oxygen”
(2) Content-wise I believe the experiments are interesting but I disagree with quite a few aspects of their interpretations. However, this may still be of interest to the field. My main concern is the superoxide detection. Authors reference to the nice paper of Adzic et al [23], this suggests superoxide at the surface of Pt in alkaline from the range from ca 0 to 0.7 V vs RHE. What I found curious is that superoxide was observed below 0.34 V by the authors, below 0.34 V there is nothing but H. Now the point is the authors have assigned some signals from the XAS to the pressence of different species, but the assignage is determined by a model (FEFF) and a fixed background, but given PtO2 and metalic Pt XAS differs vastly in backround and the fact that small Pt Nanoparticle makes the system exceptional sensitive to surface changes e.g. from coordination and oxidation I believe the authors to a higher degree should make it clear that the species assignmet is a working hypothesis.
As we described in the section 3.5, we observed that the XAS of Pt-O, or Pt-O2 is different from that of the XAS observed above 0.34 V. Therefore, with taking into account the published works of [5, 6, 23, 24, 34] in the reference of the present manuscript, it is reasonable to conclude that the XAS observed above 0.34 V originates from a superoxide anion. However, we agree with the reviewer2 that our experimental technique did not give a direct observation regarding the adsorbate. Then, we modified the way of saying about the conclusion in Line 239: “ we expect that this new adsorbate could be a superoxide anion [5, 6, 23, 24, 34]”
